# Molecular mechanisms of Bdp1 in TFIIIB assembly and RNA polymerase III transcription initiation

Jerome Gouge[1], Nicolas Guthertz[1], Kevin Kramm[2], Oleksandr Dergai[3], Guillermo Abascal-Palacios[1], Karishma Satia[1], Pascal Cousin[3], Nouria Hernandez[3], Dina Grohmann[2] & Alessandro Vannini[1]

Initiation of gene transcription by RNA polymerase (Pol) III requires the activity of TFIIIB, a complex formed by Brf1 (or Brf2), TBP (TATA-binding protein), and Bdp1. TFIIIB is required for recruitment of Pol III and to promote the transition from a closed to an open Pol III pre-initiation complex, a process dependent on the activity of the Bdp1 subunit. Here, we present a crystal structure of a Brf2–TBP–Bdp1 complex bound to DNA at 2.7 Å resolution, integrated with single-molecule FRET analysis and in vitro biochemical assays. Our study provides a structural insight on how Bdp1 is assembled into TFIIIB complexes, reveals structural and functional similarities between Bdp1 and Pol II factors TFIIA and TFIIF, and unravels essential interactions with DNA and with the upstream factor SNAPc. Furthermore, our data support the idea of a concerted mechanism involving TFIIIB and RNA polymerase III subunits for the closed to open pre-initiation complex transition.

[1] The Institute of Cancer Research, London SW7 3RP, UK. [2] Department of Biochemistry, Genetics and Microbiology, Institute of Microbiology, University of Regensburg, 93053 Regensburg, Germany. [3] Center for Integrative Genomics, Faculty of Biology and Medicine, University of Lausanne, 1015 Lausanne, Switzerland. Jerome Gouge and Nicolas Guthertz contributed equally to this work. Correspondence and requests for materials should be addressed to A.V. (email: alessandro.vannini@icr.ac.uk)

RNA polymerase III (Pol III) is the eukaryotic multisubunit DNA-dependent enzyme dedicated to the transcription of short untranslated RNAs involved in essential cellular processes, such as the entire pool of transfer RNAs, the 5S ribosomal RNA, and the spliceosomal U6 small nuclear RNA (snRNA).

Transcription initiation requires the action of the TFIIIB complex, which is both required and sufficient to recruit Pol III to the promoter and to faithfully initiate transcription in yeast[1]. The TFIIIB complex is composed of three subunits: the TATA-binding protein (TBP), TFIIB-related factor 1 (Brf1), and Bdp1[2, 3]. In vertebrates, the TFIIB-related factor II (Brf2) replaces Brf1 at type III promoters such as the U6 snRNA promoter, which, unlike the other types of Pol III promoters, are characterized by the presence of a strong TATA box and an upstream proximal sequence (PSE) recognized by the multisubunit transcription factor SNAPc[4–6] (Fig. 1a). Although in *Saccharomyces cerevisiae* and human cells the majority of Pol III promoters recognized by the Brf1-containing complex do not have a TATA box, a TATA box can recruit both Brf1-containing TFIIIB complexes and

Brf2-containing TFIIIB complexes[2, 3]. In the context of such TFIIIB complexes, TBP recognizes specifically the TATA box inducing a sharp bend of the DNA, whereas the TFIIB-related factors Brf1 and Brf2, similarly to their Pol II paralog TFIIB, recognize TATA-flanking regions and interact directly with Pol III, thus effectively bridging the polymerase to its target promoters[4, 5]. While Brf1 and Brf2 interact tightly with TBP and co-purify, the third component of the TFIIIB complex, Bdp1, is weakly associated within the TFIIIB complex in absence of DNA but contributes to the formation of an extremely stable TFIIIB–DNA complex[7–9]. Accordingly, TFIIIB complexes have been shown to act as genomic roadblocks that induce dissociation of the DNA replicative machinery at lagging strands, as well as termination of transcription derived from neighboring Pol II transcriptional units limiting pervasive transcription[10, 11]. In this context, Bdp1 appears to play an important role since Bdp1 depletion has been shown to abolish TFIIIB roadblock function[10]. Bdp1 is a transcription factor unique to the Pol III system as a Bdp1 paralog in the Pol I and Pol II system has not been identified. Bdp1 contains a highly conserved SANT domain, which

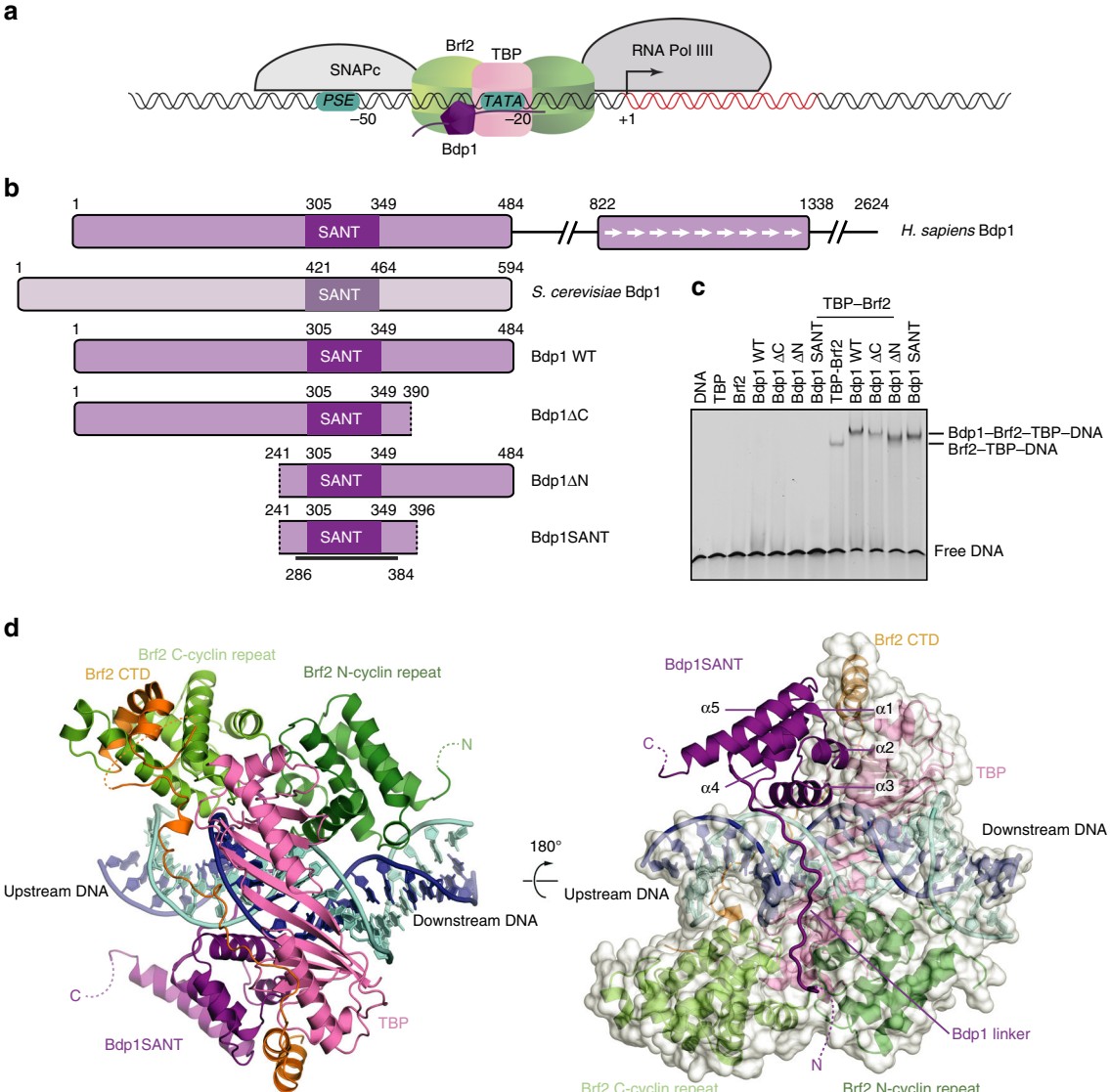

**Fig. 1** Structure of the Brf2–TBP–Bdp1–DNA complex. **a** Architecture of a RNA Pol III type III promoter. *Numbers* indicate the position of promoter elements relative to the transcriptional start site (+1). **b** Schematic overview of Bdp1 proteins and variants used in this study. The *black line* represents the region for which an atomic model could be built in the electron density. **c** EMSA showing binding of different Bdp1 constructs to Brf2–TBP–DNA complexes (*last four lanes*). **d** Two views of the Brf2–TBP–Bdp1–DNA complex. The template and non-template DNA strands are depicted in *blue* and *cyan*, respectively

**Table 1 Data collection and refinement statistics**

| | TFIIIB/DNA |
|---|---|
| *Data collection* | |
| Space group | $P2_1$ |
| Cell dimensions | |
| *a, b, c* (Å) | 91.2, 124.1, 88.6 |
| *α, β, γ* (°) | 90, 99.5, 90 |
| Resolution (Å) | 49–2.7 (2.79–2.7)[a] |
| $R_{pim}$ (%) | 0.11 (1.13) |
| $I/\sigma I$ | 5.3 (0.7) |
| Completeness (%) | 99.2 (99.7) |
| Redundancy | 9.2 (9.3) |
| CC1/2 | 0.988 (0.257) |
| *Refinement* | |
| Resolution (Å) | 49–2.7 |
| No. of reflections | 431,731 |
| $R_{work}/R_{free}$ (%) | 19.7/23.3 |
| No. of atoms | |
| Protein | 9247 |
| DNA | 2124 |
| Ligand/ion | 2 |
| Water | 252 |
| B-factors | |
| Protein | 85.1 |
| DNA | 91.5 |
| Ligand/ion | 88 |
| Water | 58.2 |
| R.m.s deviations | |
| Bond lengths (Å) | 0.01 |
| Bond angles (°) | 1.02 |

Each structure was determined from one crystal
[a]Highest resolution shell is shown in parenthesis

constitutes the major module interacting with Brf1–TBP and Brf2–TBP complexes[12–15]. The SANT domain together with the N-terminal and C-terminal flanking regions forms an extended SANT domain, which is essential for viability in yeast (Fig. 1b)[14, 16]. Bdp1 has been shown to form a complex network of interactions within the Pol III pre-initiation complex (PIC) and to play an essential role in DNA strand separation during the transition from a closed to an open form of the PIC, as transcriptionally inactive Bdp1 mutants could be rescued by pre-opening DNA templates in yeast[16, 17]. In particular, the Bdp1-extended SANT domain is required for Pol III PIC assembly and stability[16].

In the human system, several Bdp1 isoforms exist, which contain C-terminal extensions of different lengths characterized by the presence of a 55-residue repetitive motif[12]. Albeit important for regulation, the human-specific Bdp1 C-terminal extension is not strictly required for transcription, as a yeast-like human Bdp1 truncated protein is capable of faithfully initiating transcription in vitro[13]. Recently, Bdp1 has been found to be essential in promoting tumorigenicity in P53-mutated prostate cancers[18]. Despite its central role in the assembly of TFIIIB complexes and in the initial steps of Pol III transcription initiation, the molecular architecture and function of Bdp1 are still poorly understood.

Here, we describe the crystal structure of a Brf2–TBP–Bdp1 complex bound to a naturally occurring DNA promoter and, integrating the structure with single-molecule FRET and biochemical analysis, unravel the central role of Bdp1 in TFIIIB assembly and RNA Pol III PIC formation. The SANT domain of Bdp1 interacts with both TBP and DNA, while a structured linker emerging at the N terminus of the SANT domain interacts with the minor groove of the DNA. Binding of Bdp1 to Brf2–TBP–DNA complexes confers stability to TFIIIB and provides essential binding surfaces for the upstream factor SNAPc. Assembly of Bdp1 into a TFIIIB complex does not result

into DNA melting or destabilization, suggesting that Bdp1 might allosterically alter Pol III-subunit conformations during the closed to open PIC transition.

## Results

**Crystal structure of a human TFIIIB–DNA complex**. To gain insight into the structure and function of Bdp1 in the context of TFIIIB complexes, we expressed and purified to homogeneity several variants of human Bdp1 based on secondary structure predictions and homology with the yeast protein (Fig. 1b and Supplementary Fig. 1). A minimal construct encompassing the conserved Bdp1-extended SANT domain (residues 241–396, referred to as Bdp1SANT hereafter) binds to the Brf2–TBP–DNA complex with similar affinity to longer Bdp1 constructs, as shown by electrophoretic mobility shift assays (EMSAs; Fig. 1c). This suggests that the extended SANT domain represents the major anchoring point of human Bdp1 within the TFIIIB complex, in agreement with what was previously observed in yeast[19].

Bdp1SANT could be crystallized by vapor diffusion in complex with Brf2, TBP, and a DNA fragment of 25 base pairs (bp) of the human U6 promoter (referred to as TFIIIB–DNA complex hereafter), and the X-ray structure was solved at 2.7 Å by molecular replacement (Table 1 and Supplementary Fig. 2). In the TFIIIB–DNA complex (Fig. 1d), Bdp1 can be schematically divided into a compact domain constituted by the Bdp1 SANT domain (helix 1–3, residues 303–348) as well as two additional C-terminal α-helices (helix 4 and 5, residues 348–382), and an N-terminal extension that forms an ordered linker (residues 286–302). Bdp1 residues 241–285, which are predicted to form a β-strand region and are essential for promoter opening in yeast[17, 20], and 383–396 are not visible in the electron-density map, suggesting that in the absence of Pol III these regions are disordered. Comparison with a Brf2–TBP–DNA structure[4] reveals that, while the DNA path is only slightly altered, Bdp1 binding locally perturbs the geometry of the Brf2–TBP complex and results in a rearrangement of the Brf2 N-terminal and C-terminal repeats relative to TBP (Fig. 2). In particular, at the downstream TATA-flanking region the Brf2 N-terminal cyclin fold moves away from TBP, leaving Brf2 interactions with the template strand unperturbed but causing a shift of register of one nucleobase on the non-template strand (Fig. 2a and Supplementary Fig. 3). This change appears to be induced by the binding of Bdp1 to the Brf2–TBP–DNA complex, but we cannot rule out the possibility that the different crystal packing as well as the different DNA scaffold used in this study might additionally play a role. Nevertheless, these data suggest that, while the DNA template strand is held very tightly within the TFIIIB complex, the non-template strand displays a certain degree of flexibility.

Toward the end of the Bdp1 linker and at the top of the major groove, Bdp1 S293 clashes with a phosphate group of the DNA template strand, which is shifted away and stabilized by a strong hydrogen bond with Bdp1 Y291, resulting in disengagement of Brf2 Y260 at the GR element[4], a DNA sequence recognized directly by the Brf2 protein upstream the TATA box, and a movement of the entire Brf2 C-terminal cyclin fold (Fig. 2b). Thus, while recognition of the GR element is important for the initial binding of Brf2–TBP complexes at Brf2-dependent promoters, this interaction is loosened upon Bdp1 binding.

The molecular pin of Brf2, a short helical element important for the assembly and regulation of Brf2–TBP–DNA complexes[4], is unaltered in the TFIIIB structure, suggesting that binding of Bdp1 does not alter the redox-sensing properties of Brf2.

**Bdp1SANT tripartite interaction with TBP–Brf2 and the DNA**. In the TFIIIB–DNA complex, Bdp1SANT lays at the interface

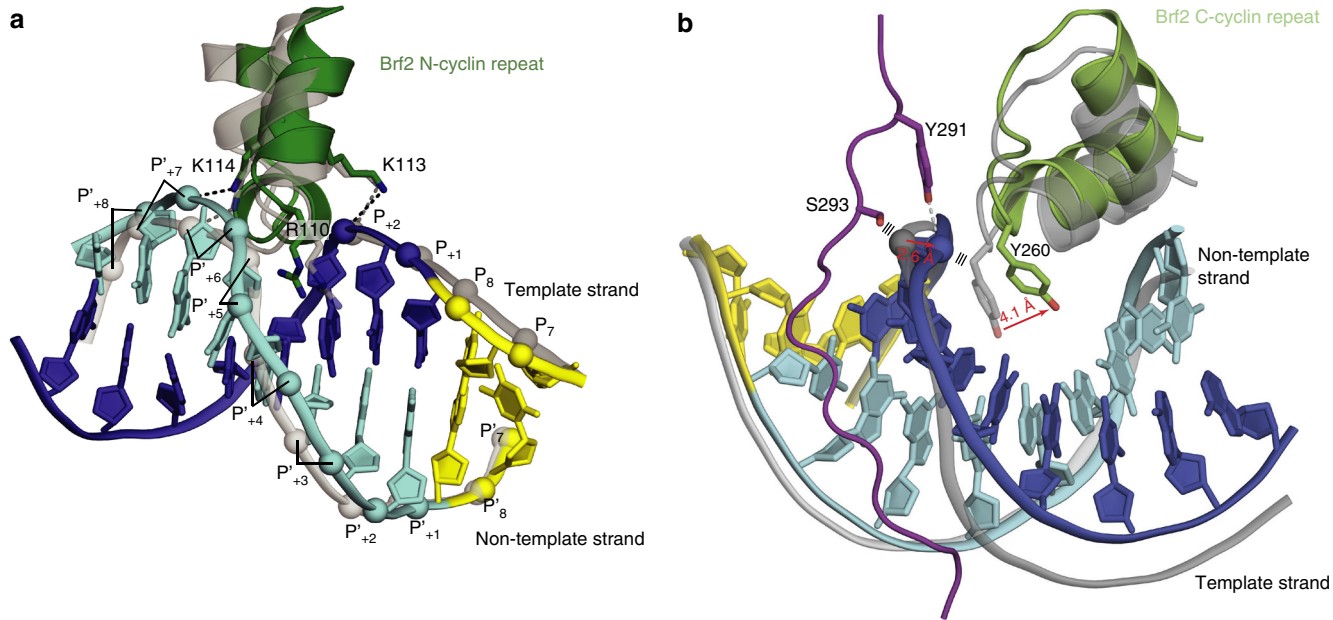

**Fig. 2** Structural rearrangement of Brf2 N-cyclin and C-cyclin repeats upon Bdp1 binding. **a** In presence of Bdp1, the Brf2 N-cyclin repeat (depicted in *green*) is rearranged when compared to the Brf2–TBP–DNA structure (PDB id:4ROC, depicted in *gray*) causing a shift of one nucleobase in the non-template strand (in *cyan*). The template strand (in *blue*) is held in place by interactions with Brf2 K113 and R110. The TATA box is depicted in *yellow*. **b** The Bdp1 linker rearranges the phosphate backbone of the DNA template strand causing the Brf2 C-cyclin repeat (depicted in *green*) to move away when compared to the Brf2–TBP–DNA structure (depicted in *gray*)

between the DNA region immediately upstream of the TATA box, the TBP N-terminal stirrup, and the Brf2 TBP anchor domain—the high-affinity TBP-binding site at the very C terminus of Brf2[4]—in agreement with recent crosslinking studies[16, 21] (Fig. 1d). The observed location overlaps with the binding site of transcription factor TFIIA in the Pol II system (Supplementary Fig. 4)[22–24]. However, in contrast to TFIIA, the Bdp1SANT establishes tight interactions not only with TBP but also with the DNA. The recognition helix (α3, residues 334–347) of Bdp1SANT contacts both sides of the major groove of the DNA, using conserved polar residues to interact with the phosphate backbone of the template and non-template DNA strands (Fig. 3a and Supplementary Fig. 3). In particular, the invariably conserved Bdp1 residue R334 forms a salt bridge with TBP residue E191, and contacts the DNA phosphate backbone of the template strand (Fig. 3a). Highlighting the importance of this interaction, a Bdp1 R334E mutation, which inverts the net polar charge, reduces Bdp1 binding to the Brf2–TBP complex, as shown by EMSAs (Fig. 3b). In agreement with this finding, the E191K mutation in human TBP and the corresponding TBP E93R mutation in yeast, both polar charge-inverting mutations, have been shown to strongly reduce TFIIIB assembly and Pol III-specific transcription[25, 26]. This suggests that the location of Bdp1 SANT domain is additionally conserved in Brf1-containing TFIIIB complexes. Impacting on recruitment of Bdp1 onto the TFIIIB complex and, consequently, on the formation of productive Pol III PICs, the Bdp1 R334E mutation substantially reduces the Pol III transcriptional output, as shown in an in vitro transcriptional assay (IVT) based on a type III Brf2-dependent promoter and recombinant factors (Fig. 3c). The residual Pol III activity suggests, however, that redundant interactions, likely involving SNAPc, additionally contribute to efficient Bdp1 recruitment at type III promoters.

**A Bdp1 linker invades the DNA minor groove.** The N-terminal region flanking the Bdp1 SANT domain, which we named the

"Bdp1 linker" (residues 286–302), adopts an extended conformation, spanning the entire length of the minor groove of the DNA and inserting in between the two Brf2 cyclin repeats (Figs. 1d and 4a). The Bdp1 linker is enriched in aromatic residues, a feature that is conserved throughout evolution (Supplementary Fig. 1). Bdp1 residues Y299 and F294 anchor the Bdp1 linker to the major groove of the DNA through aromatic-sugar interactions with the DNA backbone (Fig. 4a). Bdp1 F294 buries into a hydrophobic cavity formed by the sugar moiety of the DNA, TBP L187, and Brf2 L146 (Fig. 4a). This interaction is likely to be conserved throughout evolution, including yeast Brf1-containing TFIIIB complexes, as an aromatic residue is invariantly present in Bdp1 homologs, TBP L287 is universally conserved and Brf2 L146 is highly conserved or invariably replaced by a hydrophobic residue in yeast and human Brf1[4] (Supplementary Fig. 1).

The orientation of the Bdp1 linker relative to the SANT domain is determined by conserved polar interactions involving the Bdp1 R332 residue. R332 is a strictly evolutionarily conserved residue and is held in position by E307, another highly conserved residue, to form hydrogen bonds with two adjacent residues of the linker (a carbonyl oxygen and a conserved Ser/Thr residue; Fig. 4b). Additionally, R332 forms a hydrophobic stacking interaction with the strictly conserved residue W303.

Deletion of the Bdp1 linker (residues 288–299, Bdp1 Δlinker) results in reduced binding of Bdp1 to the Brf2–TBP–DNA complex, as shown by EMSA (Fig. 4c), and in appreciable reduction of transcriptional output in IVTs (Fig. 3c). In agreement with these findings, deletion of the Bdp1 linker region conferred lethality in yeast[14]. Further analysis revealed that deletion of the Bdp1 linker (residues 409–421) caused a defect in promoter opening on linear templates, which could only be partially rescued by pre-opening the DNA template. In contrast, deletion of the adjacent β-strand region resulted in a severe defect in promoter opening that could be fully restored by pre-opening the DNA, suggesting a major role for the β-strand region in promoter opening[27]. Thus, the Bdp1 linker is an essential region

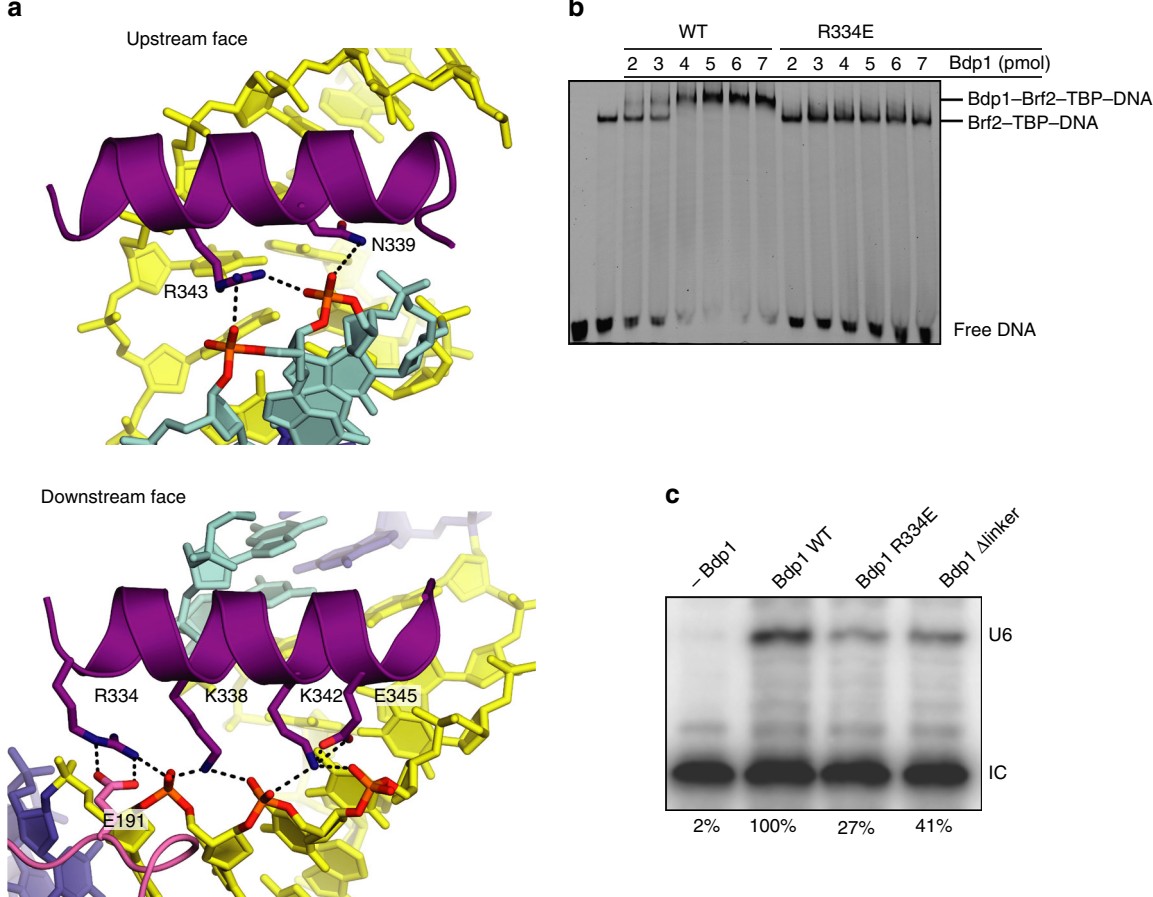

**Fig. 3** Interactions of the SANT domain of Bdp1 with DNA and TBP. **a** The recognition helix of the Bdp1 SANT domain interacts with both faces of the major grove of the DNA through conserved polar interactions. Bdp1 R334 forms a strong salt bridge with TBP E191. **b** EMSA showing reduced binding of Bdp1 R334E mutant to Brf2–TBP–DNA complexes. **c** A Bdp1 R334E mutation reduces U6 transcription in vitro. IC is the internal loading control. The values at the *bottom* represent the % of the normalized intensities relative to the wild-type Bdp1 sample. The IC band was used for normalization

which plays a role in promoter opening, likely by optimally positioning the adjacent β-strand region in the Pol III PIC, but also by having an impact on Bdp1 recruitment onto TFIIIB complexes.

**Dynamics of TFIIIB–DNA binding.** In order to understand the dynamics of the system and the relative contribution of TBP, Brf2, and the Bdp1-extended SANT domain in the assembly of a functional TFIIIB complex on the DNA, we employed single-molecule FRET to follow the bending of the U6 promoter DNA upon binding of the TFIIIB components (Fig. 5, Table 2, Supplementary Fig. 5 and Supplementary Table 1). As observed for Pol II promoters, binding of TBP to the TATA box induces the initial bending of the DNA[28], while Brf2 and Bdp1 in isolation have no effect on DNA bending (Fig. 5b, Table 2, Supplementary Fig. 5 and Supplementary Table 1). Human TBP induces a single bent state of the U6 promoter (Fig. 5a), and the FRET efficiency oscillates between high and low values, suggesting a rapid association of TBP with the DNA and subsequent rapid dissociation (lifetime of the TBP–DNA complex = 0.29 s, Supplementary Fig. 6a). This observation is in agreement with a previous study that reported very short lifetimes (2–20 s) for human TBP in complex with a canonical RNA Pol II promoter[29]. However, the situation is very different in yeast, where the TBP–DNA complex can adopt two conformations that differ in their bending angles distinguished by an extremely high stability of the complex (lifetime of 12 min)[28].

Upon addition of TBP and Brf2 to the promoter DNA, molecules showed that stable high FRET and washing of the measuring chamber with only buffer did not result in loss of the high FRET states (Fig. 5b). In contrast to what observed for the binary TBP–DNA complex, the bend DNA molecules in the Brf2–TBP–DNA complex showed that stable FRET and no dynamics could be detected, a further indication that Brf2 stabilizes the TBP–DNA interaction. Analysis of confocal single-molecule measurements exploiting the FRET signal in solution estimated a lifetime of slightly over 6 min (381 s) for the Brf2–TBP–DNA complex (Fig. 5b and Supplementary Fig. 6d). It is noteworthy that the dynamic interaction of human TBP with the U6 promoter DNA and the stabilization of the TBP–DNA complex by the TFIIB-like factor Brf2 is reminiscent of the highly dynamic interaction of archaeal initiation factors TBP and TFB with promoter DNA[28].

Addition of Bdp1SANT to the Brf2–TBP–DNA complex has no effect on the DNA-bending angle but increases the estimated lifetime of the complex to ~9 min (547 s, Supplementary Figs. 5 and 6d and Supplementary Table 1). In support of this finding, an EMSA competition assay, using an unlabeled competitor DNA, confirms that the addition of Bdp1SANT results in an increased stability of the full TFIIIB complex (Supplementary Fig. 7), as shown by relatively high level of binding of TFIIIB complexes to labeled DNA even in presence of 200-fold excess of unlabeled competitor. Thus, in agreement with the crystal structure, addition of Bdp1SANT to Brf2–TBP–DNA complexes does not

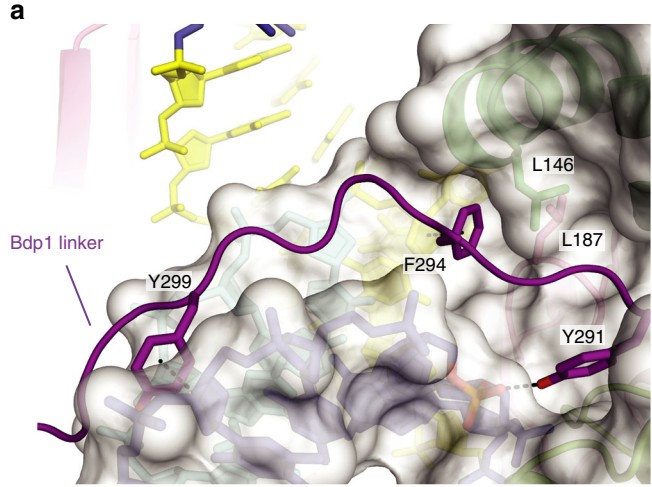

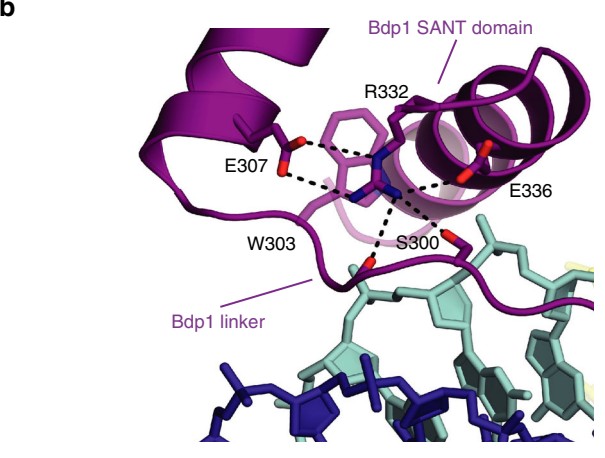

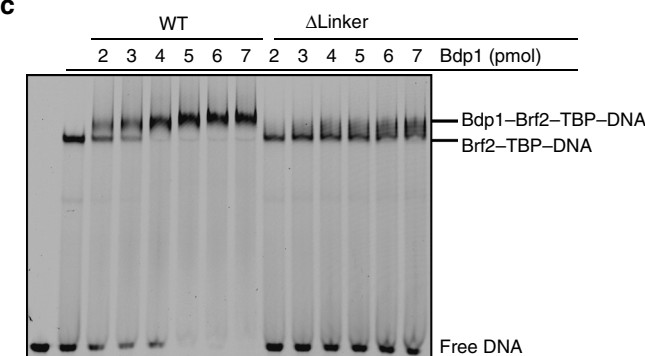

**Fig. 4** Characterization of the Bdp1 linker. **a** The Bdp1 linker invades the DNA minor groove and interacts with the DNA through aromatic residues. *Color scheme* as in Fig. 1c; the TATA box is depicted in *yellow*. **b** The Bdp1 linker interacts with the SANT domain throughout conserved polar residues. *Color scheme* as in Fig. 1c. **c** EMSA showing reduced binding of Bdp1 Δlinker mutant to Brf2–TBP–DNA complexes.

alter the bending state of the DNA, suggesting that Bdp1 binds to the stable pre-bent Brf2–TBP–DNA complex resulting in extremely stable complexes bound to the DNA.

**Bdp1 N-terminal regions and C-terminal regions**. Albeit efficiently recruited onto TFIIIB complexes (Fig. 1c), Bdp1SANT is

virtually inactive in IVTs (Fig. 6a). A Bdp1 construct lacking the N-terminal region (Bdp1-ΔN) is also inactive in IVTs, while a construct lacking the C-terminal region (Bdp1-ΔC) shows reduced activity (Fig. 6a). This result suggests that Bdp1 N-terminal regions and C-terminal regions play an important role in driving faithful Pol III transcription. As noted before, one possibility is the involvement of Bdp1 terminal extensions in binding to the essential upstream factor SNAPc. Bdp1 interacts strongly with SNAPc in absence of DNA, as shown in pull-down assays (Fig. 6b), as well as in presence of DNA, as shown by EMSA (Fig. 6c). Binding of Bdp1-ΔN to SNAPc is reduced both in the presence and absence of DNA, while Bdp1-ΔC is still able to bind to SNAPc at appreciable levels (Fig. 6b, c). Removal of both the N-terminal and C-terminal regions, which is the aforementioned Bdp1SANT variant, abolishes binding to SNAPc (Fig. 6b, c). These results closely mirror the results obtained in IVTs, suggesting that the terminal regions of Bdp1 are capable of binding to SNAPc and that these interactions are essential to support RNA Pol III transcription.

**A model of the Pol III PIC**. During Pol III transcription initiation, Bdp1 stimulates the transition from a closed to an open PIC, a process that occurs without ATP hydrolysis[30]. The structure presented here reveals that, in the context of an isolated TFIIIB complex, Bdp1 does not induce DNA melting by stabilizing flipped nucleobases, as observed in bacterial σ factors[31, 32]. To further investigate the role of Bdp1 in promoter opening, we combined the TFIIIB structure presented here with the available atomic models of Pol III[33] and of a human Pol II-PIC[23] and generated a model of a Pol III-PIC (Fig. 7). The model reveals that the Bdp1 SANT domain is distant from the core of the polymerase, analogously to TFIIA in RNA Pol II[23, 24], contributing to the stability of the PIC[23, 24]. On the other hand, after exiting the Brf2 cyclin folds the Bdp1 linker points toward a surface formed between the protrusion and the wall domains, an analogous location as observed for the Tfg2 linker of TFIIF in Pol II PICs (Fig. 7), which provides additional stability to TFIIB on the wall of Pol II[23, 24]. As a result, the predicted N-terminal β-strand Bdp1 segment, which is essential for promoter opening[17], could be located in the proximity of the Pol III protrusion, similarly to the TFIIF Tfg2 linker and Tfg1 arm, which add β-strands to the Pol II protrusion, resulting in the stabilization of the DNA bubble. These findings suggest that Bdp1 might stimulate DNA strand separation through an allosteric mechanism involving RNA Pol III subunits, as previously hypothesized[17].

**Discussion**

The structure of the TFIIIB–DNA complex reveals how Bdp1 is recruited onto a Brf2–TBP–DNA complex through essential interactions, which are conserved throughout evolution. Bdp1 is organized in a compact extended SANT domain, which represents the major Bdp1 anchor point to the Brf2–TBP–DNA complex, and an N-terminal linker that invades the DNA minor groove (Fig. 1c). The SANT domain binds in an analogous location as observed for TFIIA, while the Bdp1 linker, and the adjacent β-strand segment, might bind similarly to the Tfg2 linker of TFIIF, according to our model of a RNA Pol III PIC. Thus, albeit structurally unrelated, Bdp1 exploits comparable binding surfaces and mechanisms utilized by distinct transcription factors in the Pol II system.

Furthermore, Bdp1 strongly interacts with the upstream DNA, on both the major and the minor groves, thus contributing to the high stability of TFIIIB complexes on the DNA[10, 15] (Figs. 1d, 3a and 4a). Bdp1 binds efficiently only to pre-bent Brf2–TBP–DNA

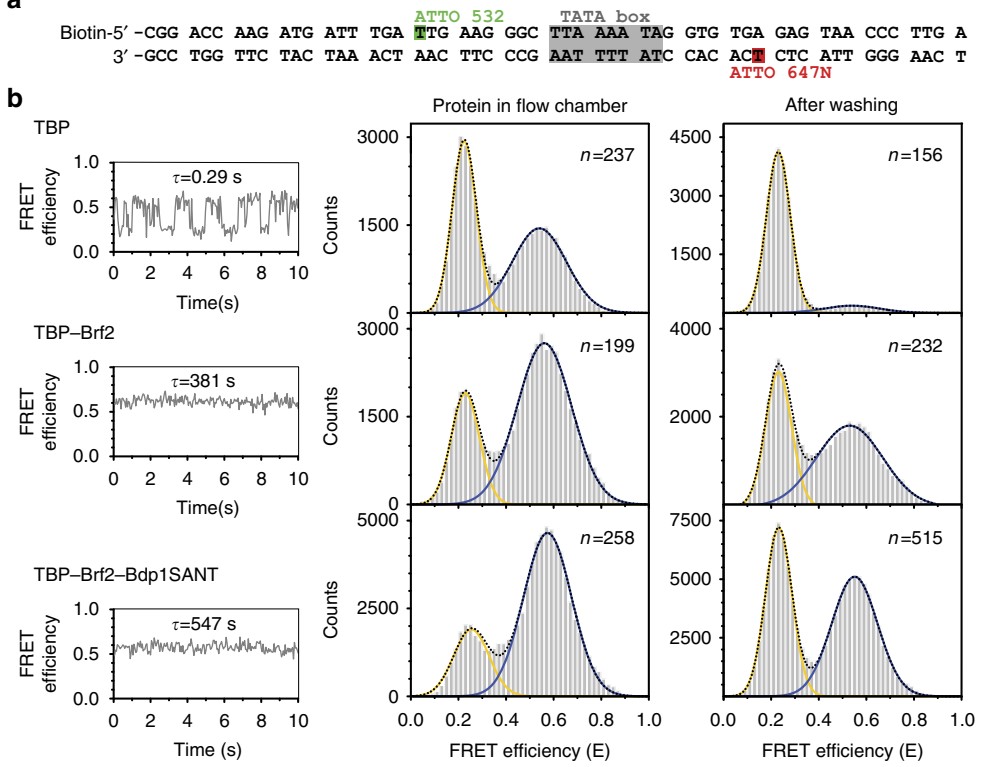

**Fig. 5** Influence of transcription factors on promoter DNA conformation. **a** Single-molecule FRET measurements were carried out using a synthetic U6 snRNA promoter. FRET between an internal donor (ATTO 532) and acceptor (ATTO 647 N) dye was used as a readout to follow transcription initiation factor-induced bending of the promoter DNA. **b** Measurements on immobilized molecules were conducted once with the proteins present in the flow chamber and after a washing step that removed freely diffusing proteins. A representative intensity–time trace for each measurement is shown depicting the dynamic association and dissociation of TBP (10 nM) with and from the DNA with a complex lifetime of $\tau = 0.29$ s. In contrast, addition of Brf2 results in a stable complex characterized by a single high FRET state with a complex lifetime of 381 s. Addition of Bdp1SANT increased the lifetime of the high FRET state to 547 s. FRET efficiency histograms show the distribution between low FRET (free DNA) and high FRET states (bend DNA). The total number of molecules used for each histogram is given as n. Each histogram was fitted with a single or double Gaussian distribution to determine the mean FRET efficiencies (see Table 2). Each measurement was carried out at least three times

complexes (Fig. 5), with the recognition helix of the Bdp1 SANT domain contacting both sides of the major groove (Fig. 3a). Binding of Bdp1 causes a subtle rearrangement of both the N-terminal and C-terminal cyclin repeats of Brf2, resulting in a loosened interaction with the upstream segment of DNA around the GR element. In contrast, the template strand in the downstream region around the TD motif[4], a DNA sequence downstream the TATA box that is specifically recognized by Brf2 residues R110 and A108, is held tightly (Fig. 2a, b). Thus, the TFIIIB complex displays a certain degree of plasticity, allowing structural rearrangements during the initial stages of transcription, without compromising the interactions with the template strand in proximity of the transcription start site. In this respect, the situation is somewhat reminiscent of the bacterial σ factors, which strongly interact with both template and non-template strands, pre-organizing the DNA to engage the Pol active center[32]. However, in contrast to bacterial σ factors, TFIIIB does not directly induce DNA melting by stabilizing flipped-out nucleobases, suggesting an indirect role of TFIIIB in promoter opening. Accordingly, our model of a Pol III PIC reveals structural and functional analogies between domains of Bdp1 and Pol II general transcription factors within the Pol II PIC, which have been recently shown to spontaneously cause DNA opening[24].

The molecular details of how Bdp1 promotes DNA melting will require further structural investigations in the context of an intact Pol III PIC.

## Methods

**Recombinant-protein expression and purification.** TBP (159–339) and Brf2 (62–419) have been expressed in Rosetta (DE3)pLysS *Escherichia coli* cells. TBP and the TBP–Brf2 complex were both expressed overnight at 20 °C with 1 mM IPTG (Isopropyl β-D-1-thiogalactopyranoside) in a TB medium. Full-length Brf2 was expressed at 30 °C for 4 h with 1 mM IPTG. Cells were grown at 37 °C until the optical density reached 0.6 (A600), and then the temperature was reduced and IPTG was added 30 min after. Cells were harvested by centrifugation at 4000×*g* (20 min at 4 °C) and the pellets stored at −80 °C. For crystallization, pellets of Brf2 co-expressed with TBP were re-suspended in 750 mM NaCl, 10 mM imidazole, 50 mM HEPES pH 7.9, 10% glycerol, 5 mM β-mercaptoethanol (buffer A1), supplemented with protease inhibitors tablets (Pierce), and DNase I (20 mg/mL). Cells were lysed through sonication and clarified by centrifugation (14,000×*g* for 45 min at 4 °C). The supernatant was applied onto a 5 mL HisTrap HP column (GE Healthcare) and bound proteins were washed with 50 mL of buffer A1 supplemented with 50 mM imidazole, and then eluted with buffer B1 (buffer A1 containing 300 mM imidazole). The salt concentration was reduced to 350 mM by diluting the sample with buffer H1 (50 mM HEPES pH 7.9, 10% glycerol, 5 mM dithiothreitol (DTT)). The protein samples were loaded onto a 5 mL HiTrap Heparin HP column (GE Healthcare) and eluted with a 375 mM to 1.5 M NaCl gradient in 10 column volumes. At this stage, the Brf2–TBP complex was dissociating and the subsequent steps were performed to purify Brf2 in isolation. The fractions containing the protein of interest were pooled and the N-terminal His-tag was cleaved overnight at 4 °C with the 3 C protease. Imidazole (20 mM) was added to the sample before loading onto a 5 mL HisTrap HP column. The flowthrough was concentrated and applied onto a Superdex 200 26/600 column (GE Healthcare). The size exclusion chromatography was performed in a buffer containing 500 mM ammonium acetate, 50 mM HEPES pH 7.9, 10% glycerol, and 2 mM DTT (gel filtration buffer). The fractions containing Brf2 were pooled, concentrated, flash-frozen, and stored at −80 °C. For full-length Brf2, the same protocol was used, with the exception that the tag was not cleaved and the second nickel column step was omitted.

His-TBP pellets were re-suspended in buffer A2 (A1 buffer containing 500 mM NaCl) supplemented with protease inhibitor tablets (Pierce) and DNase I (20 mg/mL).

**Table 2 FRET efficiencies (E) and percentage of dynamic molecules for the single-molecule FRET promoter-bending essay**

| | Protein in flow chamber | Standard error | $R^2$ | After washing | Standard error | $R^2$ |
|---|---|---|---|---|---|---|
| *TBP* | | | | | | |
| $E_{low}$ $(x_c \pm \sigma)$ | 0.22 ± 0.05 | 0.001 | 0.9907 | 0.23 ± 0.05 | 0.0003 | 0.9988 |
| $E_{high}$ $(x_c \pm \sigma)$ | 0.54 ± 0.11 | 0.003 | | 0.53 ± 0.12 | | |
| dyn. mol. (%) | 46 | | | 6 | | |
| *TBP–Brf2* | | | | | | |
| $E_{low}$ $(x_c \pm \sigma)$ | 0.23 ± 0.06 | 0.002 | 0.9911 | 0.23 ± 0.05 | 0.001 | 0.9893 |
| $E_{high}$ $(x_c \pm \sigma)$ | 0.56 ± 0.12 | 0.002 | | 0.53 ± 0.14 | 0.003 | |
| dyn. mol. (%) | 12 | | | 5 | | |
| *TBP–Brf2–Bdp1$_{SANT}$* | | | | | | |
| $E_{low}$ $(x_c \pm \sigma)$ | 0.25 ± 0.07 | 0.004 | 0.9869 | 0.23 ± 0.06 | 0.001 | 0.9970 |
| $E_{high}$ $(x_c \pm \sigma)$ | 0.58 ± 0.10 | 0.002 | | 0.55 ± 0.10 | 0.001 | |
| dyn. mol. (%) | 8 | | | 5 | | |

E is shown as the mean ($x_c$) with standard deviation ($\sigma$) and standard error of a single or double Gaussian fit with the coefficient of determination ($R^2$). Samples were first measured with the proteins in the flow chamber and again after washing away unbound proteins with buffer

Cell lysis and clarification were performed as mentioned above and the supernatant applied on a 5 mL HisTrap HP column (GE Healthcare). The bound proteins were washed with 50 mL of buffer A2 supplemented with 50 mM imidazole, and then eluted with buffer B2 (similar to buffer B1 but with 500 mM NaCl). The fractions containing His-TBP were pooled and loaded onto a 5 mL HiTrap Heparin HP column (GE Healthcare). The protein was eluted with 500 mM to 2 M NaCl gradient in 10 column volumes. The tag was cleaved overnight at 4 °C by addition of the 3 C protease. The reaction was then adjusted to 20 mM imidazole and loaded onto a HisTrap HP column. The flowthrough was concentrated and loaded onto a Superdex 200 16/600 column (GE Healthcare) in gel filtration buffer. After elution, the protein was concentrated, flash–frozen, and stored at −80 °C.

Bdp1-WT (residues 1–484) and Bdp1-ΔC were cloned into pOPINE vector (uncleavable C-terminal His-tag), while Bdp1SANT and Bdp1-ΔN were cloned into pOPINF vector (cleavable N-terminal His-tag). Proteins were expressed in *E. coli* BL21-CodonPlus (DE3)-RIL cells (Agilent Technologies) in TB media. Expression was induced when cells reached OD$_{600}$ = 2 with 0.5 mM IPTG for 16 h at 18 °C.

Bdp1 proteins were purified from clarified lysate through consecutive Ni-NTA (Qiagen) affinity chromatography. For crystallization of Bdp1SANT, the affinity tag was removed by cleavage with 3 C protease for 16 h at 4 °C with a protease/protein molar ratio of 1:50. Further purification of recombinant proteins was achieved through HiTrap Heparin HP (5 mL) chromatography (GE Healthcare) and size-exclusion chromatography (HiLoad 16/600 Superdex 200, GE Healthcare) in 50 mM HEPES pH 7.8, 500 mM Ammonium Acetate, 10% glycerol, and 2 mM DTT. Protein samples were concentrated to 0.15 mM with VivaSpin 10,000 Daltons molecular weight cut-off centrifugal filter units (Generon) and were stored at −80 °C after flash-freezing in liquid nitrogen.

SNAPcΔ is a complex comprising subunit SNAPc190Δ (residues 1–516) with a N-terminal Strep-tag, SNAPc50 with a N-terminal His tag, SNAPc43, and SNAPc19. The genes were cloned in pACEBAC1 and expressed in insect cells. Hi5 cells were harvested after 4 days of infection by centrifugation at 250×*g* for 10 min at 4 °C and the pellets stored at −80 °C. The re-suspension buffer is composed of 750 mM NaCl, 10 mM imidazole, 50 mM HEPES pH 7.9, 10% glycerol, 5 mM β-mercaptoethanol (buffer A0), supplemented with protease inhibitor tablets (Pierce) and DNase I (20 µg/mL). Cells were lysed through sonication and were clarified by centrifugation (14,000×*g* for 45 min at 4 °C). The supernatant was applied onto a 5 mL HiTrap HP column (GE Healthcare) and bound proteins were washed with 50 mL of buffer A1 (500 mM NaCl, 50 mM HEPES pH 7.9, 50 mM imidazole, 5 mM β-mercaptoethanol, and 10 mM O-Phospho-L-serine). The complex was then eluted with a buffer containing 500 mM NaCl, 50 mM HEPES pH 7.9, 300 mM imidazole, and 5 mM β-mercaptoethanol. The salt concentration was reduced to 250 mM by diluting the sample with buffer H1 (50 mM HEPES pH 7.9, 10% glycerol, and 1 mM TCEP). The sample was loaded onto a 5 mL HiTrap Heparin HP column (GE Healthcare) and eluted with a 250 mM to 1.25 M NaCl gradient in 10 column volumes. The fractions containing the protein of interest were pooled. The N-terminal His-tags were cleaved with the 3 C protease and the sample was dephosphorylated with the lambda phosphatase during an overnight incubation at 4 °C. Imidazole (30 mM) was added to the sample before loading onto a 5 mL HiTrap HP column. The flowthrough was concentrated and applied onto a Superdex 200 26/600 column (GE Healthcare). The size exclusion chromatography was performed in a buffer containing 100 mM NaCl, 50 mM HEPES pH 7.9, 10% glycerol, and 1 mM TCEP.

**Oligonucleotides**. All oligonucleotides for EMSAs and crystallography were purchased from Integrated DNA Technologies. Each oligonucleotide (single strand) was re-suspended in 50 mM Tris pH 8.0, 5 mM MgCl$_2$, and 1 mM EDTA, and was mixed with the complementary strand in an equimolar ratio, heated at 95 °C for 3 min, and then allowed to cool down to room temperature overnight. Fluorescently labeled oligonucleotides for single-molecule FRET analysis were

purchased from IBA (Göttingen). Single-strand oligonucleotides were re-suspended in 10 mM Tris-HCl pH 8.0, 50 mM NaCl EDTA, and mixed with the complementary strand in an equimolar ratio, heated at 95 °C for 3 min, and then allowed to cool down to room temperature overnight.

**Crystallization, data collection, and processing**. DNA sequences used for crystallization are the following: 5′-GGTCACACCTATTTTAAGCCCTTCAAC-3′ (template strand) and 5′-TTGAAGGGCTTAAAATAGGTGTGAC-3′ (non-template strand). Overhangs were used to stabilize the crystal packing. The complexes were assembled at a final concentration of 60 µM with an equimolar ratio of TBP and Brf2, 1.1 molar excess of Bdp1SANT, and 1.2 excess of the double-stranded DNA (dsDNA). Crystals were grown by mixing 1 µl of the complexes and 1 µl of the crystallization solution (10–16% PEG 8000, 150–200 mM KCl, 100 mM MES pH 6.5, 10–15 mM MgCl$_2$, and 1 mM TCEP) in hanging drop plates. Seeds were introduced after 16 h of equilibration. After 3 days, crystals were cryo-protected in a mix of 50% paraffin/50% paratone and flash-frozen in liquid nitrogen. X-ray diffraction data consist of two data sets recorded on the same crystal after translating the beam to a different region of the crystals. Both were collected at 0.919761 Å, with 0.02 s exposition and 0.1° oscillation, for a total of 7200 images on beamline ID29 at the ESRF (France). The data were indexed and scaled with XDS and XSCALE[34], and merged with AIMLESS[35]. The data were processed using CC$_{1/2}$ and completeness as cutoff criterion[36].

**Structure determination and refinement**. The structure containing the SANT domain was solved by molecular replacement using the TBP–Brf2–DNA structure (PDB id: 4ROC) as a search model in PHASER[37]. A rigid body refinement of each of the Brf2 cyclin repeats was carried out independently, and the resulting density maps clearly showed α-helices corresponding to the Bdp1 SANT domain. Iterative manual building was performed with COOT[38], and BUSTER-TNT (version 2.10.1) was used for refinement[39]. The quality of the final structure was assessed with MolProbity[40]. Data collection and refinement statistics are shown in Table 1. The final model has excellent stereochemistry (Ramachandran allowed 99.3%). The figures were prepared with PyMol (version 1.8.4.0 Schrödinger LLC). The alignments were performed with Clustal Omega[41] and prepared with ESprit[42].

**Electrophoretic mobility shift assay**. The oligonucleotides used for the interaction of Bdp1 on the TBP–Brf2 complex are Cy-5-labeled and based on the U6_2 sequence: 5′-Cy5-ATTTGATTGAAGGGCTTAAAATAGGTGTGA-CAGTAACC-3′ and 5′-GGTTACTGTCACACCTATTTTAAGCCCTTCAATC AAAT-3′. The complexes were assembled in a 20 µl reaction volume in a buffer containing 500 mM ammonium acetate, 50 mM HEPES pH 7.9, 10% glycerol, and 1 mM TCEP. The complexes were assembled by mixing 1.5 pmol of the dsDNA with 5 pmol of TBP (159–339), 7 pmol of Brf2 (62–419), and 10 pmol of Bdp1 construct, except stated differently. The complexes were then incubated at 21 °C for 20 min. Prior to loading the sample in the EMSA, the reactions were diluted twofold in a buffer containing 100 mM ammonium acetate, 50 mM HEPES pH 7.9, 40% glycerol, and 1 mM TCEP. The binding reactions were resolved on a 4.5% polyacrylamide (37.5:1 acrylamide/bis-acrylamide, 10% glycerol, Tris borate EDTA 1×) gel in 1× Tris-borate EDTA running buffer at 40 mA.

The complexes for the competition assays in Supplementary Fig. 7 were assembled as described above, but using 15 pmol of the Bdp1SANT domain. The complexes were assembled in presence of the labeled DNA and incubated for 20 min before adding the unlabeled competitor (same sequence). The complexes were incubated for an hour before loading on the gel.

The oligonucleotides used to monitor the interaction between Bdp1 and SNAPcΔ are also based on the U6_2 sequence but contain 7 bp upstream and downstream of the PSE (specified in italic): 5′-CCATAAG*TTATCCTAACCAAAA*

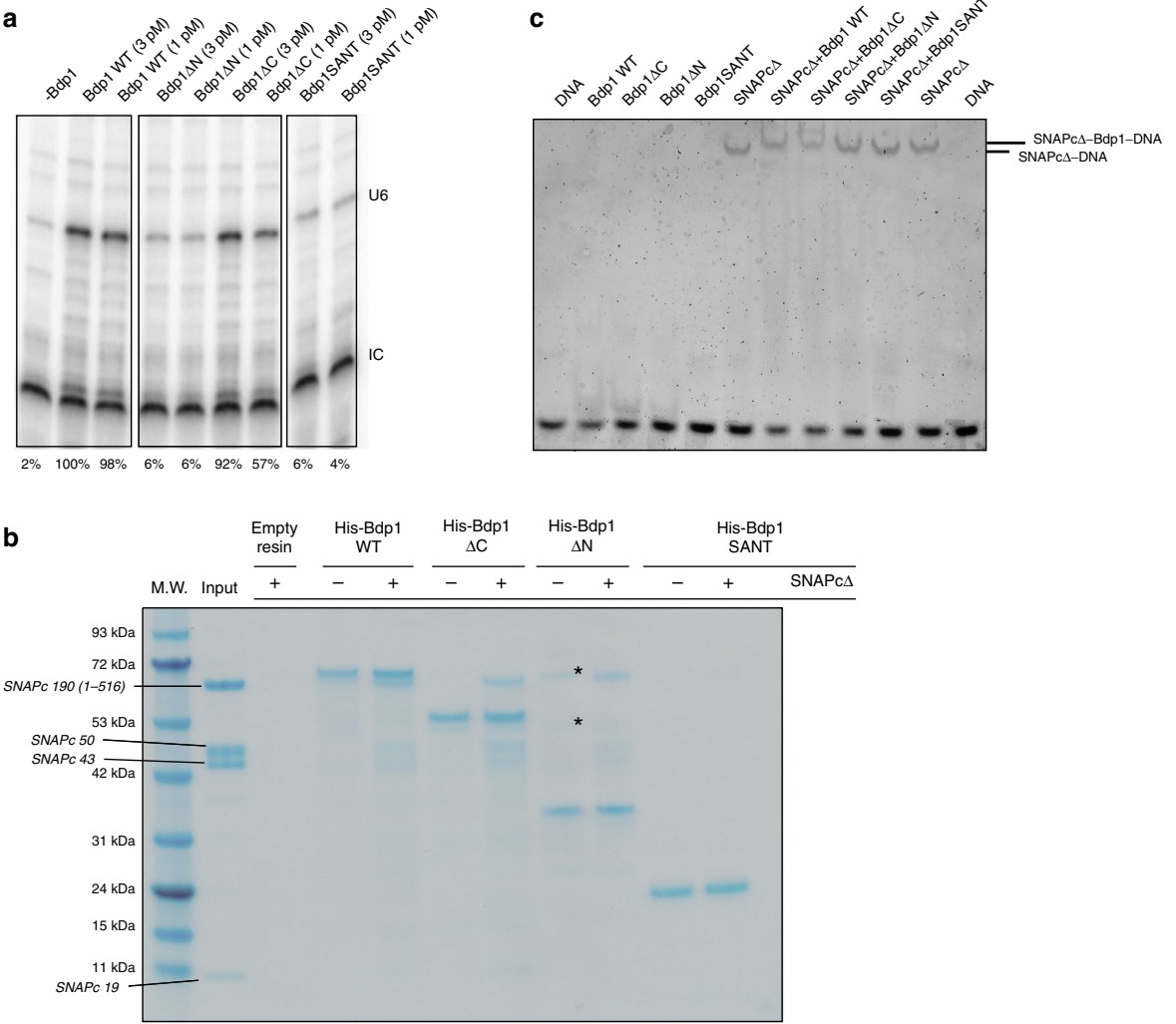

**Fig. 6** Functional characterization of the Bdp1 N-terminal and C-terminal regions. **a** Deletion of N-terminal and C-terminal regions of Bdp1 reduces U6 transcription in vitro. IC is the internal loading control. The values at the *bottom* represent the % of the normalized intensities relative to the wild-type Bdp1 sample. The IC band was used for normalization. **b** Pull-down assay using immobilized Bdp1 constructs showing that deletion of N-terminal and C-terminal regions of Bdp1 reduces binding of Bdp1 to SNAPc in absence of the DNA. The composition of the SNAPcΔ is shown as a control in the *input lane* with the SNAPcΔ subunits labeled aside the molecular weight (M.W.) *standards lane*. *Indicates uncharacterized contaminants in the Bdp1-ΔN preparation. **c** EMSA showing that deletion of N-terminal and C-terminal regions of Bdp1 reduces binding of Bdp1 to SNAPc in presence of the DNA

*GAT*GATTTGA-3′ and 5′-TCAAATC*ATCTTTTGGTTAGGATAAC*TTATGG-3′. The complexes were assembled in a 20 μl reaction volume in a buffer containing 500 mM NaCl, 50 mM HEPES pH 7.9, 20% glycerol, and 1 mM TCEP. The complexes were assembled by mixing 10 pmol of the dsDNA with 10 pmol of SNAPcΔ and 15 pmol of the different Bdp1 constructs. The complexes were then incubated at 21 °C for 20 min. The binding reactions were resolved on a 4.5% polyacrylamide (37.5:1 acrylamide/bis-acrylamide, 10% glycerol, Tris borate EDTA 1×) gel in 1× Tris- borate EDTA running buffer at 40 mA.

The gels were scanned with a Typhoon FLA 9500 and band visualization was carried out with ImageQuanT TL version 8.1. For Supplementary Fig. 7, the mean and standard deviation were calculated on the basis of four and two independent experiments for TBP–Brf2 and TBP–Brf2–Bdp1SANT, respectively. The data were plotted with GraphPad Prism 6. EMSA reactions were loaded on a sodium dodecyl sulphate-polyacrylamide gel electrophoresis (SDS-PAGE) gel, ensuring protein qualitative and quantitative controls.

**In vitro transcription assay**. In vitro transcription assays were performed as described[43] with minor modifications. Tagged pol III complex, recombinant SNAPc, and recombinant TBP, Brf2, and Bdp1 were mixed with 250 ng of poly (dG-dC) · poly(dG-dC) and 80 ng of pU6/Hae/RA.2 construct in 2% glycerol, 5 mM HEPES (pH 7.9), 200 mM ammonium acetate, 5 mM MgCl₂, 0.025 mM EDTA, 1 mM DTT, and protease inhibitors tablets (Pierce).

**Single-molecule FRET measurements**. Single-molecule FRET measurements on immobilized DNA–protein complexes were carried out in custom-built flow

chambers based on fused silica slides passivated with polyethylene glycol (PEG). Flow chambers were prepared and assembled as described before[28]. Briefly, quartz slides were cleaned in peroxymonosulfuric acid. Afterward, the slides and cover-slips were sequentially sonicated with water, Helmanex, water, and methanol. The slides and coverslips were functionalized with N-[3-(trimethoxysilyl)propyl]ethy-lenediamine in acidic methanol, rinsed with methanol and water, and dried at 37 °C for 1 h. The slides and coverslips were coated with a methoxi-PEG/biotin-PEG mixture for 2 h at 21 °C, and were washed with water and finally dried at 37 °C for 2 h. The slides and coverslips were stored in evacuated tubes at −20 °C. To prepare the flow chambers, a slide and coverslip were warmed to room temperature before opening. A piece of parafilm with cut out flow channels was sandwiched between the slide and coverslip and fused together by heating the assembled flow chamber to 90 °C. Tubes for sample injection were fixed with epoxy resin. The flow chambers were cleaned with 2-propanol.

For measurements the flow chamber was incubated with 0.1 mg/mL NeutrAvidin (Pierce) in TBS (125 mM Tris-HCl pH 8, 150 mM NaCl) for 5 min and washed with 500 μL T78 (100 mM Tris-HCl pH 7.8, 60 mM KCl, 5 mM MgCl₂, 0.5 mg/mL bovine serum albumin (BSA), 1% (v/v) glycerol). Afterward, the chamber wash flushed with the pre-annealed biotinylated promoter DNA (10 pM in TBS; Fig. 5a) for 5 s and washed with 500 μL T78.

For complex formation, protein aliquots were thawed on ice and diluted stepwise to a concentration of 1 nM (TBP) or 0.1 nM (Brf2, Bdp1) with pre-chilled T78. Afterward, 250 μL oxygen-scavenging buffer (80 mM Tris-HCl pH 7.8, 48 mM KCl, 4 mM MgCl₂, 0.4 mg/mL BSA, 0.8% (w/v) glycerol, 1% (v/v) glucose, 2 mM TROLOX, 7.5 U/mL glucose oxidase type VII (Sigma-Aldrich), and 1 kU/mL

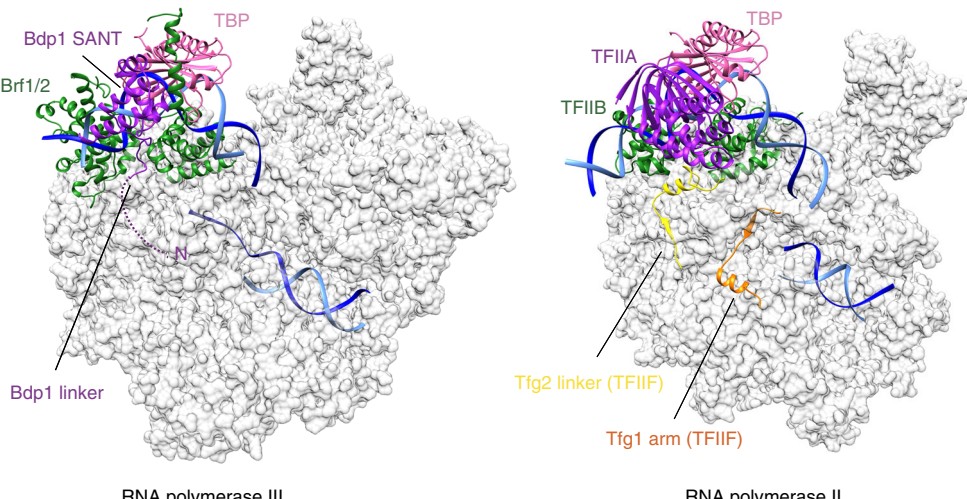

**Fig. 7** Architecture of the RNA Pol III PIC. Model of a Pol III PIC generated using the TFIIIB–DNA complex and the atomic model of elongating Pol III[33] reveals structural similarities with the yeast and human Pol II PIC[23, 24]. *Color coding* is the same as in Fig. 1d

catalase from bovine liver (Sigma Aldrich)), which contained the respective transcription factors at a final concentration of 10 nM (TBP) or 1 nM (Brf2, Bdp1) was flushed into the chamber and incubated for 5 min. A first set of single-molecule FRET measurements was carried out at 20 °C in the presence of proteins in the buffer before washing the chamber with 250 μL T78 buffer to remove unbound proteins. Subsequently, the chamber was flushed with 250 μL oxygen-scavenging buffer and incubated for 5 min before a second set of single-molecule FRET measurements on stable complexes was recorded. A home-built prism-type total internal reflection set-up based on a Leica DMi8 inverse research microscope was used for all measurements[44]. A 532 nm solid state laser (Coherent OBIS, 30 mW power) and a 637 nm diode laser (Coherent OBIS, clean-up filter ZET 635/10, Chroma, 50 mW power) employing alternating laser excitation (Multistream, Cairn Research) were used to excite the fluorophores[45].

A Leica HC PL Apo ×63 numerical aperture (NA) 1.20 water-immersion objective was used to collect fluorescence. Fluorescence was split by wavelength with a dichroic mirror (HC BS 640, Semrock) into two detection channels that were further filtered with a 582/75 bandpass filter (Brightline HC, Semrock) in the green channel, and a 635 nm long-pass filter (LP Edge Basic, Semrock) in the red detection channel. A single EMCCD camera (Andor IXon Ultra 897, EM-gain 20, framerate 40 Hz, 400 frames) in a dual-view configuration (TripleSplit, Cairn Research) was used to record both detection channels.

The videos were analyzed employing the iSMS software[46] using the programs' default settings. Molecule spots were detected using a threshold of 100 for ATTO532 and ATTO647N spots. FRET efficiencies were calculated as proximity ratios from fluorescence intensity time traces, which were corrected for background fluorescence using the average intensity of all pixels with a 2 pixel distance to the molecule spot. For FRET efficiency histograms (E-histograms), all frames of selected traces with an S-value of 0.25–0.55 were used. The resulting FRET efficiency histograms shown are accumulated data from at least three independent measurements. The FRET efficiency histograms were fitted with either a single or double Gaussian fit to calculate the mean FRET values and standard errors. For dwell time histograms, traces showing dynamic switching between FRET states were fitted with the vbFRET algorithm limited to two states[47]. All states with a FRET efficiency within the full-width at half-maximum of the high FRET population were used to calculate the dwell time histogram. The histograms of at least three independent experiments were normalized and fitted with a mono-exponential decay function to calculate the mean dwell time in the high FRET state (TBP bound to DNA). In order to determine the lifetime of the long-lived TBP–Brf2 and TBP–Brf2–Bdp1SANT complexes, we performed confocal single-molecule fluorescence measurements in solution. Prior to sample loading, chambers (Cellview slide, Greiner Bio-One) were passivated with Tris-HCl pH 8 with 2 mg/mL BSA for 10 min and washed once with T78. For complex formation, 20 pM fluorescently labeled promoter DNA and 50 nM TBP–Brf2 or TBP–Brf2–Bdp1SANT were incubated for 15 min at room temperature in T78 and 1 mM DTT. Single-molecule fluorescence of diffusing complexes was detected with a MicroTime 200 confocal microscope (Picoquant) equipped with pulsed laser diodes (532 nm: LDH-P-FA-530B; 636 nm: LDH-D-C-640 PicoQuant/cleanup filter: ZET635/10; Chroma). The fluorophores were excited at 20 μW using pulsed interleaved excitation. Emitted fluorescence was collected using a 1.2 NA, ×60 microscope objective (UplanSApo ×60/1.20 W; Olympus) and a 50 μm confocal pinhole was used. A dichroic mirror (T635lpxr, Chroma) separated donor and acceptor fluorescence. Additional bandpass filters (donor: ff01-582/64, Chroma; acceptor: H690/70, Chroma) completed spectral separation of the sample fluorescence. After 5 min of measuring time, 50 μg/mL heparin was added, which

traps the dissociated proteins avoiding re-binding of the proteins to the fluorescently labeled DNA. Data analysis was performed with the SymPhoTime 64 software (Picoquant). Intensity–time traces were calculated using a 1 ms binning. Fluorescent molecule bursts were detected using a threshold of 25 counts. An E–S histogram was calculated for every 5 min interval of the trace (t = 0 corresponding to the 5 min prior to heparin addition). The histograms were normalized to the total number of bursts and were fitted using a single or double Gaussian fit. The ratio of the high-FRET peak area vs. the total fit area was plotted over time and fitted with a mono-exponential decay to determine the average lifetime.

**Pull downs**. To study the interaction between the different Bdp1 constructs and SNAPcΔ, 50 μg of Bdp1-tagged constructs were loaded onto His SpinTrap columns (GE Healthcare) in presence or in absence of 100 μg of SNAPcΔ, in a buffer containing 250 mM NaCl, 50 mM HEPES pH 7.9, 10 mM imidazole, 10% glycerol, and 2 mM β-mercaptoethanol in a total reaction volume of 300 μL. The incubation was performed at 4 °C for 2 h. The beads were extensively washed with the same buffer supplemented with 50 mM imidazole. The proteins were eluted with 75 μl of the binding buffer containing 300 mM imidazole, boiled, and analyzed on a sodium dodecyl sulphate-polyacrylamide gel electrophoresis.

**Data availability**. Atomic coordinates and structure factors for the reported crystal structure have been deposited in the Protein Data Bank under accession code 5N9G. All other relevant data are available from the corresponding authors.

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

## Acknowledgements

We thank N. Cronin at the Institute of Cancer Research for help with the crystallization set-up. We thank the staff at beamline ID29 of the European Synchrotron Radiation Facility (France) for help with data collection. D.G. acknowledges funding by the DFG (SFB960/TP A7). P.C., O.D. and N.H. were supported by the University of Lausanne and by Swiss National Science Foundation (SNSF) grants 31003B_149904 and 31003A_169233. G.A.-P. is a recipient of a Marie Sklodowska-Curie Intra-European Fellowship (EU project 655238). A.V. acknowledges the support of the Career Development Faculty Program of the ICR. This work was supported by a Biotechnology and Biological Sciences Research Council (BBSRC) new-investigator award (BB/K014390/1), a Cancer Research UK Programme Foundation (CR-UK C47547/A21536), and a Wellcome Trust Investigator Award (200818/Z/16/Z) to A.V.

## Author contributions

J.G. and N.G. carried out purification of protein/nucleic acid complexes and performed biochemical experiments, including EMSAs and pull downs. N.G. prepared TFIIIB–DNA crystals. J.G. and N.G. collected crystallographic data and solved the TFIIIB–DNA structures. K.S. and G.A.-P. helped with cloning and purification. K.K. and D.G. performed single-molecule FRET and analyzed the results. P.C., O.D., and N.H. performed in vitro transcription assays and analyzed the results. A.V. designed and supervised research, and prepared the manuscript with contributions from all authors.

## Additional information

**Competing interests:** The authors declare no competing financial interests.

