## [Peer Review file · Nature Communications]

Reviewers' comments:

Reviewer #1 (Remarks to the Author):

The authors report the x-ray structure of a human TFIIIB-U6 DNA complex containing TBP, Brf2 and the Bdp1 extended SANT domain. This is an important structure and reveals a number of interesting properties of the complex and the roles of the three subunits in complex formation. The structure reveals how the Bdp1 SANT domain and linker interact with TBP, DNA and Brf2. The structure explains how Bdp1 contacting both TBP and DNA stabilizes the TFIIIB complex. FRET experiments revealed that Bdp1 does not affect the bending of the TBP-Brf2-DNA complex but presumably stabilizes the preformed complex. The authors show that regions of Bdp1 outside of the SANT region cooperate with the regulatory factor SNAPc and this is nicely demonstrated with a combination of binding and transcription assays. Its very interesting that Bdp1 binds in a similar location on TBP as TFIIA and the linker may be in an analogous position to part of TFIIF. Finally, the authors use their structure to propose a model for the Pol III PIC that is sure to stimulate further investigations into the workings of the Pol III basal factors. I recommend the manuscript be accepted after addressing the comments below:

1) pg 5: please indicate the residues encompassing the Bdp1 linker the first time it is mentioned. The first mention of this (residues 286-302) in the current manuscript is pg 7.

2) pg 6: Define the GR element and the Brf2 anchor domain. Also, define the TD motif on pg 12.

3) Fig 6 and pg 10: The description of Fig 6b is not sufficient to determine how the experiment was done and what band = what protein. Rewrite figure legend and label proteins in the figure. What was added to the IN lane (snapc delta?). What are the extra bands in the IN lane? Also, the figure is not convincing for the author's claim that the Bdp1 N-term region is more important for snap binding compared to the C-term region.

4) pg 12, paragraph 1: It's an overstatement to say that the Bdb1 linker binds similarly to the Tfg2 linker of TFIIF as this is only speculation at this point. Make sure statements like this are clearly labeled as models.

Reviewer #2 (Remarks to the Author):

Vannini et al. describe the crystal structure of human TFIIIB at 2.7 Å resolution that comprises a ternary complex of TBP, Brf2 and a Bdp1 construct containing mainly the SANT domain bound to a 25 bp DNA duplex of the human U6 promoter. The manuscript presents the intricate structure of the TFIIIB-DNA complex, where the Bdp1 SANT domain and a preceding linker provide additional stabilization to the Brf2-TBP-DNA complex through several contacts between SANT domain and DNA backbone, but also between the N-terminal Bdp1 linker and the minor groove of the DNA as well as the two Brf2 cyclin repeats. The TFIIIB-DNA complex structure extends previous work on the TBP-Brf2-DNA complex from the same group as it also shows the role of Bdp1 in stabilizing the DNA-binding complex. Structural information is complemented by EMSA and in vitro transcription experiments. In addition, the authors use single-molecule FRET to demonstrate the dynamics of TFIIIB binding. These experiments demonstrate rapid association and dissociation of TBP with DNA, whereas addition of Brf2 dramatically increases the stability of the Brf2-TBP-DNA complex, while further addition of Bdp1 does not lead to additional stabilization. Finally, the authors demonstrate the role of the Bdp1 N- and C-terminal extensions in Pol III transcription and suggest a model how the N-terminal β -strand segment (not present in the crystal structure) could contribute to DNA melting similar as proposed for TFIIF.

Overall, this comprehensive study combines challenging structural biology with biochemistry and

single-molecule FRET experiments. The crystallographic analysis is sound and the data presented are of high quality. I recommend publication in Nature Communications once the following points have been addressed:

1. The presented TFIIIB-DNA complex structure and previous literature suggest considerable stabilization of TFIIIB by Bdp1, whereas the FRET experiments do not show additional stabilization of the DNA-bound complex by Bdp1. Can the author detect additional stability of the TFIIIB-DNA complex (TBP-BRF2-Bdp1-DNA) compared to the TBP-BRF2-DNA complex using thermal unfolding for example in a thermofluor assay? If successful, this assay could demonstrate how much for example the N-terminal Bdp1 linker stabilizes the TFIIIB complex.

2. The similar positions of the Bdp1 linker and the TFIIIF linker and arm are interesting. The authors should extend their observations by comparing the Pol II and Pol III PIC architectures in greater detail. Is there any indication that indeed Bdp1 also adds β -strands to the Pol III protrusion as observed for TFIIIF.

Minor points:

-In the EMSA experiments the authors should indicate the position of free DNA, TBP-Brf2-DNA complex and TBP-BRF2-Bdp1-DNA complex.

-On page 12: "the adjacent β -strand segment binds similarly to the Tfg2 linker of TFIIIF". Because the β -strand segment is not included in the crystallisation construct, the wording should be changed to make clear that this is a hypothesis.

-In Figure 4 it would be informative to show electron density for the linker.

-Page 14: DNase concentration in the lysis buffer (20 mg/ml) should be corrected.

Reviewer #3 (Remarks to the Author):

The authors report on the structure of TFIIIB, complex of Brf2, TBP and Bdp1, providing important insight into how the complex is formed and exactly how it might support the formation of a transcriptionally competent TFIIIB-Pol III complex. They provide single-molecule FRET data to inform on the stability of the various complexes formed by subsets of the different TFIIIB proteins. I will limit my comments here to these experiments.

- The authors should insert an introductory cartoon in Figure 1 that explains to the reader the juxtaposition between Brf2, TBP, Bdp1, and Pol III on the promotor. To the uninitiated reader, the first few paragraphs are difficult to follow without a visual aid.

- The authors provide a very quantitative description of the observed FRET levels in the single-molecule experiments, but leave out critical information. For example, in Table 2, they should indicate which FRET level is observed in the (sometimes large) population of molecules that do not show dynamics. Also, they should provide example traces for all the conditions depicted in Supp Figure 5.

- It is remarkable that the binding lifetime of the TBP (3 seconds) is so different compared to that obtained from the yeast TBP (12 minutes; Ref 27). The authors should briefly comment on this difference.

- The authors comment on how "Addition of Brf2 fully shifts the equilibrium towards the bent state...". However, inspection of the histograms in Figure 5b suggest the presence of the unbent state. From the presented data, it is not clear whether these correspond to DNA molecules that haven't properly assembled the complex, or whether these correspond to the low FRET state after the complex dissociates. In other words, the authors have to present more kinetic information on these experiments.

- Similarly, it's a lost opportunity not to measure and report the binding lifetimes of each of the complexes that show stable FRET in Figure 5. A simple reduction of excitation power and a concomitant decrease in time resolution would allow the smFRET to be stretched out long enough to characterize the full kinetics. The authors should add this information.

- From the methods section, it appears that the authors have used an alternating-laser excitation approach to gather stoichiometry information. The authors mention that they only analyse the FRET signals of molecules that have a stoichiometry parameter of 0.25-0.55. Why not use a window that is symmetrically centered around 0.5?

- The authors mention that they fit dwell times with a single-exponential function. They should show the histograms and the fits.

We are extremely pleased to note that the manuscript was very well received by the 3 reviewers. We would like to thank them for their constructive comments that helped us improving the manuscript. As described below, we have addressed all the points of the reviewers by textual and/or graphical revision and added new experiments (EMSA and FRET) that show higher stability of the Bdp1-Brf2-TBP/DNA complex (when compared to Brf2-TBP/DNA), as requested by reviewers#2 and #3. Additionally, we provided additional information to better evaluate the FRET data as requested by reviewer#3.

In particular here follow a point-to-point rebuttal to address the reviewers' comments:

REVIEWER#1:

1) pg 5: please indicate the residues encompassing the Bdp1 linker the first time it is mentioned. The first mention of this (residues 286-302) in the current manuscript is pg 7.

Thanks for pointing out this inconsistency, we now mention the residue range at page 5 where the Bdp1 linker is first mentioned.

2) pg 6: Define the GR element and the Brf2 anchor domain. Also, define the TD motif on pg 12.

We have now defined the GR element and TD motif as well as the TBP anchor domain in the text.

3) Fig 6 and pg 10: The description of Fig 6b is not sufficient to determine how the experiment was done and what band = what protein. Rewrite figure legend and label proteins in the figure. What was added to the IN lane (snapc delta?). What are the extra bands in the IN lane? Also, the figure is not convincing for the author's claim that the Bdp1 N-term region is more important for snap binding compared to the C-term region.

We agree with the reviewer that this figure (and the legend associated) were not clear. We have re-labeled the figure 6b, Indeed the extra bands in the input lane are part of snapc delta complex, which is a multi-subunit complex. This now should be more clear. We have also expanded the figure legend and we have removed our claim in the main text that the Brf1 N-term region is more important for snap binding (compared to the C-term region).

4) pg 12, paragraph 1: It's an overstatement to say that the Bdp1 linker binds similarly to the Tfg2 linker of TFIIIF as this is only speculation at this point. Make sure statements like this are clearly labeled as models.

*We agree with the reviewer and changed the sentence into "while the Bdp1 linker, and the adjacent β -strand segment, **might** bind similarly to the Tfg2 linker of TFIIIF, **according to our model of a RNA Pol III PIC.**", which makes it absolutely clear to the reader that this evidence is based only on a model.*

REVIEWER#2:

1) The presented TFIIIB-DNA complex structure and previous literature suggest considerable stabilization of TFIIIB by Bdp1, whereas the FRET experiments do not show additional stabilization of the DNA-bound complex by Bdp1. Can the author detect additional stability of the TFIIIB-DNA complex (TBP-BRF2-Bdp1-DNA) compared to the TBP-BRF2-DNA complex using thermal unfolding for example in a thermofluor assay? If successful, this assay could demonstrate how much for example the N-terminal Bdp1 linker stabilizes the TFIIIB complex.

Thanks to the reviewer for asking to further look into this aspect. A thermofluor assay is likely to be too complicated to interpret in light of the multi-subunit nature of our complex. However, in the revised manuscript we have now added an EMSA competition assay (Supplementary Figure 7) and new single-molecule FRET measurements (Figure 5 and Supplementary Figure 5 and 6, measuring longer dwell times, see comments from reviewer#3) that clearly shows how Bdp1 confers higher stability and dwell time to the TFIIIB complex bound to DNA, in line with previous literature.

2) The similar positions of the Bdp1 linker and the TFIIIF linker and arm are interesting. The authors should extend their observations by comparing the Pol II and Pol III PIC architectures in greater detail. Is there any indication that indeed Bdp1 also adds β -strands to the Pol III protrusion as observed for TFIIIF.

As mentioned by reviewer#1, at this stage it might be too speculative to draw mechanistic parallels between the two systems. We prefer to simply highlight how the location of these factors are similar, underlying that similar surfaces are exploited during mechanisms in promoter opening/open complex stabilization.

3) Minor points:

-In the EMSA experiments the authors should indicate the position of free DNA, TBP-Brf2-DNA complex and TBP-BRF2-Bdp1-DNA complex.

We have now added the labels on our EMSAs.

-On page 12: "the adjacent β -strand segment binds similarly to the Tfg2 linker of TFIIIF". Because the β -strand segment is not included in the crystallization construct, the wording should be changed to make clear that this is a hypothesis.

We agree and we have now changed this, see comments 4 of reviewer#1.

-In Figure 4 it would be informative to show electron density for the linker.

As electron density is only for specialists, we would prefer to show it only in supplemental Fig2, leaving the clarity of Fig4 untouched.

-Page 14: DNase concentration in the lysis buffer (20 mg/ml) should be corrected.

Thank you for spotting this, we have corrected it now to the correct value (20 μ g/ml).

REVIEWER#3:

1) The authors should insert an introductory cartoon in Figure 1 that explains to the reader the juxtaposition between Brf2, TBP, Bdp1, and Pol III on the promoter. To the uninitiated reader, the first few paragraphs are difficult to follow without a visual aid.

We have now added a schematic in Figure 1, which will help readers in following the description of type III promoters.

2) The authors provide a very quantitative description of the observed FRET levels in the single-molecule experiments, but leave out critical information. For example, in Table 2, they should indicate which FRET level is observed in the (sometimes large) population of molecules that do not show dynamics. Also, they should provide example traces for all the conditions depicted in Supp Figure 5.

In the revised version of the manuscript, we added this information as well as sample traces requested by the reviewer in Figure 5 and Supplementary Figure 5 and 6.

3) It is remarkable that the binding lifetime of the TBP (3 seconds) is so different compared to that obtained from the yeast TBP (12 minutes; Ref 27). The authors should briefly comment on this difference.

Please note that in this study we not only used TBP from a different organism but also used the U6 snRNA promoter (a RNA polymerase III specific promoter) and not a prototypical RNAPII transcription promoter (H2B promoter) used in our earlier study (Gietl et al, NAR, 2014). The U6 promoter differs in the TATA-box sequence, which most likely contributes to the difference in complex lifetime. Single-molecule FRET studies with human TBP were performed earlier by the Kugel group who also detected dwell times in the range of 2- 20s for the human TBP-promoter DNA interaction using a consensus TATA-DNA. Following the suggestion of the reviewer, we added a sentence to comment on this in the revised version of the manuscript: "This observation is in agreement with a previous study that reported very short lifetimes (2-20 seconds) for human TBP in complex with a canonical RNA Pol II promoter."

4) The authors comment on how "Addition of Brf2 fully shifts the equilibrium towards the bent state...". However, inspection of the histograms in Figure 5b suggest the presence of the unbent state. From the presented data, it is not clear whether these correspond to DNA molecules that haven't properly assembled the complex, or whether these correspond to the low FRET state after the complex dissociates. In other words, the authors have to present more kinetic information on these experiments.

We thank the reviewer for pointing out to us that we explained the result in a confusing way. Hence, we rephrased this section to clarify this passage and to avoid confusion: "Addition of Brf2 results in a stable Brf2-TBP/DNA complex characterized by a mean FRET efficiency, which remained at comparable levels even after washing the measuring chamber with only buffer (Fig. 5b). In contrast to what has been observed for the binary TBP/DNA complex, the bent DNA molecules in the Brf2-TBP/DNA complex showed stable FRET and significantly reduced dynamics could be detected, a further indication that Brf2 stabilizes the TBP-DNA interaction. Analysis of confocal single-molecule measurements exploiting the FRET signal in solution estimated a lifetime of approximately 6 minutes for the Brf2-TBP/DNA complex". Please see our response at the comment below regarding the kinetic information on these experiments.

5) Similarly, it's a lost opportunity not to measure and report the binding lifetimes of each of the complexes that show stable FRET in Figure 5. A simple reduction of excitation power and a concomitant decrease in time resolution would allow the smFRET to be stretched out long enough to characterize the full kinetics. The authors should add this information.

We thank the reviewer for suggesting experimental conditions that might help in determining the complex lifetime. However, due to the high stability of the TBP/Brf2 complex we were not able to measure the lifetime with a TIRF based approach. Instead, we established experimental conditions suitable for confocal single-molecule measurements in solution. Using a similar strategy as in Gietl

et al. 2014, [Nucleic Acid Research] we were able to determine a lifetime of 381 seconds for the TBP/Brf2/DNA complex and of 547 seconds for Bdp1-SANT/Brf2/TBP/DNA complex.

6) From the methods section, it appears that the authors have used an alternating-laser excitation approach to gather stoichiometry information. The authors mention that they only analyse the FRET signals of molecules that have a stoichiometry parameter of 0.25-0.55. Why not use a window that is symmetrically centered around 0.5?

Please note that due to different laser intensities used for excitation (donor: 30 mW; acceptor: 50 mW) the S-value is shifted to lower values as compared to equal excitation powers. Hence, we chose the molecules that centre on an S value of 0.4. Please find an exemplary E-S-histogram to illustrate this:

7) The authors mention that they fit dwell times with a single-exponential function. They should show the histograms and the fits.

In the revised version of the manuscript, we included the histograms and fits of the dwell time analysis.

REVIEWERS' COMMENTS:

Reviewer #1 (Remarks to the Author):

The authors have done an excellent job in revising the manuscript in response to the reviewer comments. This work, showing structural biochemical characterization of the TFIIB complex, will have a high impact in the transcription field.

Minor point: pg 4, line 58 has a typo: delete "with Bdp1"

Reviewer #2 (Remarks to the Author):

The authors have well addressed the different issues raised by the three reviewers. I now recommend publication in Nature Communications.

Reviewer #3 (Remarks to the Author):

The authors have addressed all issues and concerns. The result is a very nice manuscript that will be appreciated by Nature Communications' readership.